# Leveraging Generative Models for Unsupervised Alignment of Neural Time Series Data

**Ayesha Vermani[†], Il Memming Park[†], Josue Nassar[‡]**
[†] Champalimaud Centre for the Unknown, Champalimaud Foundation, Portugal
[‡] RyvivyR, USA
`{ayesha.vermani, memming.park}@research.fchampalimaud.org`
`josue.nassar@ryvivyr.com`

## Abstract

Large scale inference models are widely used in neuroscience to extract latent representations from high-dimensional neural recordings. Due to the statistical heterogeneities between sessions and animals, a new model is trained from scratch to infer the underlying dynamics for each new dataset. This is computationally expensive and does not fully leverage all the available data. Moreover, as these models get more complex, they can be challenging to train. In parallel, it is becoming common to use pre-trained models in the machine learning community for few shot and transfer learning. One major hurdle that prevents the re-use of generative models in neuroscience is the complex spatio-temporal structure of neural dynamics within and across animals. Interestingly, the underlying dynamics identified from different datasets on the same task are qualitatively similar. In this work, we exploit this observation and propose a *source-free* and *unsupervised* alignment approach that utilizes the learnt dynamics and enables the re-use of trained generative models. We validate our approach on simulations and show the efficacy of the alignment on neural recordings from the motor cortex obtained during a reaching task.

## 1 Introduction

With advancements in recording techniques, we have access to a large number of simultaneously recorded neurons, exhibiting complex spatio-temporal activity. Consequently, significant efforts have been dedicated to the development of computational models that can infer the underlying structure from these recordings (Linderman et al., 2017; Pandarinath et al., 2018; Duncker et al., 2019; Schimel et al., 2022; Dowling et al., 2023). The progress in deep generative models, such as variational autoencoders (VAEs) (Kingma & Welling, 2013) and sequential variational autoencoders (Bowman et al., 2015; Hafner et al., 2019), has further contributed to a proliferation of these latent variable models for neuroscience. These models are trained to extract the latent dynamical process – typically confined to a low-dimensional manifold – that drives the high-dimensional neural or behavioral observations.

Despite the abundance of latent variable models for neural data, there are some issues that prevent their widespread adoption by the experimental community. Firstly, training large models can be data intensive; although the number of simultaneously recorded neurons continues to increase, the number of trials a subject can perform during a single experimental session is still limited (Williams & Linderman, 2021). Furthermore, there is a growing interest in studying naturalistic behaviors in the field, where trial boundaries are ill-defined and trial repetitions are few, if any (Rosenberg et al., 2021; Kennedy, 2022; Minkowicz et al., 2023). Secondly, training deep neural networks is computationally expensive and can pose several challenges. This is partly attributed to the complex relationship between training process and hyperparameter optimzation, which can considerably impact the model's performance.

In parallel, the use of pre-trained models has led to significant breakthroughs in natural language processing and computer vision (Girshick et al., 2014; Rasmy et al., 2021). These are driven by the empirical observation that model re-use is highly data efficient and achieves comparable per-

formance as a model trained from scratch, with only a fraction of the data (Goyal et al., 2019). Moreover, re-using a pre-trained model allows us to circumvent challenges associated with training a model from scratch. Recent evidence also suggests that pre-trained models are fairly generalizable and can be fine-tuned to perform a variety of tasks, even across domains (Parisi et al., 2022).

Inspired by the empirical success of pre-trained models in machine learning and the recent interest in training large models for neuroscience (Azabou et al., 2023; Ye et al., 2023), we investigate the case of using pre-trained sequential VAEs (seqVAEs) for neural time series data. seqVAEs have been widely successful at inferring the underlying latent dynamics from high-dimensional neural time series data. However, due to statistical heterogeneities across datasets, arising from disparities in the number and tuning properties of recorded neurons, differences in recording modalities, etc., pre-trained seqVAEs cannot be re-used directly on new recordings. A potential approach to tackle this problem is by learning an alignment that transforms the new dataset such that it is statistically similar to the data used to train the seqVAE. Previous approaches for learning an alignment between neural datasets require access to the original data used to train the model and/or the existence of paired samples between datasets (Degenhart et al., 2020; Chen et al., 2021; Williams et al., 2021; Duong et al., 2023; Wang et al., 2023). The paired samples are commonly constructed by arbitrarily pairing stimulus-conditioned neural activity across the datasets. This entirely ignores trial-to-trial variability and cannot be applied to naturalistic task settings. Moreover, many of these methods do not explicitly model the temporal structure of data which can lead to suboptimal learning of the alignment (Wang et al., 2023).

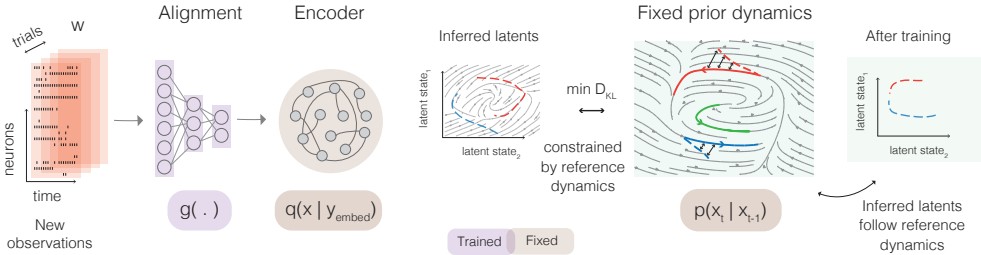

Figure 1: **Schematic of unsupervised alignment scheme**. We train a sequential VAE on some reference observations to learn an encoder, along with the underlying low-dimensional dynamics. Given new observations $y_{new}$ generated from the same dynamical process, we learn a function **g** that tranforms and implicitly aligns them to the reference, allowing for the re-use of the pre-trained model. The latent trajectories inferred after transforming the new observations, $\mathbf{g}(y_{new})$, are constrained by the learnt latent dynamics.

In this work, we propose a simple unsupervised method for aligning neural time series that facilitates the re-use of pre-trained seqVAEs. Our approach stems from the observation that learning to perform inference on new recordings using a pre-trained seqVAE implicitly results in learning an alignment between datasets. Moreover, our approach leverages the similarity in temporal dynamics across datasets to learn the alignment, as optimizing for inference in this framework encourages the inferred latents to be close to the learned dynamics (see Figure 1). As our proposed method is unsupervised, it does not require the availability of paired samples nor access to the original training data, making it highly flexible and easy to use. We empirically validate our method on synthetic experiments and test it on neural recordings obtained from the primary motor cortex (M1) of two monkeys during a center out reaching task (Dyer et al., 2017).

The main contributions of this paper are summarized as follows: **(1)** We propose a novel unsupervised method for implicit alignment of sequence data with low-dimensional dynamics that enables the re-use of trained generative models. **(2)** For the linear and Gaussian setting, we analytically demonstrate that the proposed approach recovers an alignment that is close to the optimal. **(3)** On synthetic and real data, we provide empirical evidence that the proposed method outperforms other methods. The corresponding code is available at `https://github.com/ayeshav/align-seqvae`.

## 2 RELATED WORK

There has been a large body of work on aligning neural datasets for applications such as computing metrics between neural representations (Williams et al., 2021; Duong et al., 2023); enabling the use of decoders across sessions (Sussillo et al., 2016b; Farshchian et al., 2019; Gallego et al., 2020; Degenhart et al., 2020; Ma et al., 2023; Wang et al., 2023), across animals (Herrero-Vidal et al., 2021; Chen et al., 2021), and even across species (Rizzoglio et al., 2022). One popular alignment approach minimizes the error between the original and the aligned dataset, using methods such as Canonical Correlation analysis(CCA) (Gallego et al., 2020; Rizzoglio et al., 2022) or Procrustes (Degenhart et al., 2020; Duong et al., 2023). However, they require access to the original dataset **and** require the existence of a one-to-one correspondence between the two datasets. Moreover, they don't leverage the spatio-temporal structure in neural time series. A related approach works by minimizing a divergence between the original and aligned dataset, either explicitly (Dyer et al., 2017; Karpowicz et al., 2022; Wang et al., 2023), or implicitly via generative adversarial networks (Farshchian et al., 2019; Ma et al., 2023).

The two most similar approaches to the proposed method are NoMAD (Karpowicz et al., 2022) and ERDiff (Wang et al., 2023). In NoMAD, LFADS (Sussillo et al., 2016a)—a popular seqVAE for neural time series—is first trained on the original dataset. New datasets are aligned by maximizing the log-likelihood of data and minimizing the KLD between the distribution of latent states of the original and aligned dataset, where both are assumed to be Gaussian. Crucially, since LFADS models the latent dynamics to be deterministic, the log-likelihood accounts for the spatio-temporal structure of data. While similar to the proposed method, we make no assumption on the distribution of latent states nor do we require the statistics of the original dataset. In ERDiff, a seqVAE is trained along with a spatio-temporal diffusion model that approximates the latent distribution of the original dataset. Given new data, ERDiff optimizes the alignment function to produce latent states from the pre-trained encoder that are likely under this distribution. Although this is similar in spirit to the proposed approach, there are significant differences. Namely, ERDiff requires training a spatio-temporal diffusion model, along with a seqVAE, on the source dataset to perform alignment on new data. This incurs additional overhead as re-using or sharing the pre-trained seqVAE necessitates training a diffusion model. Moreover, it does not use the learned latent dynamics for aligning as it only re-uses the encoder, instead relying on spatio-temporal transformer blocks to capture the spatio-temporal structure in the data. In contrast, our approach is considerably simpler as it only requires training an inexpensive alignment function that feeds into the encoder. Moreover, we explicitly consider the spatio-temporal structure by using the pre-trained latent dynamics.

## 3 BACKGROUND

### 3.1 SEQUENTIAL VARIATIONAL AUTOENCODER

In this work, we focus on learning an alignment between datasets that enables the re-use of state-space models (SSMs), a class of latent variable models for spatio-temporal data. Let $x_t \in \mathcal{X} \subseteq \mathbb{R}^{d_x}$ and $y_t \in \mathcal{Y} \subseteq \mathbb{R}^{d_y}$ be the low-dimensional latent state, and the observation at time $t$, respectively. A SSM can be described as follows,

$$x_t \mid x_{t-1} \sim p_\theta(x_t \mid x_{t-1}), \tag{1}$$

$$y_t \mid x_t \sim p_\phi(y_t \mid x_t), \tag{2}$$

where equation 1 is the latent dynamics distribution, parameterized by $\theta$, that describes the temporal evolution of the latent state, $x_t$; equation 2 is the likelihood distribution, parameterized by $\phi$, that maps the low-dimensional latent state to the high-dimensional observation, $y_t$. While there are many choices for the parametric form of equation 1, we follow standard practice (Krishnan et al., 2015; Hafner et al., 2019) and parameterize it as $p_\theta(x_t \mid x_{t-1}) = \mathcal{N}(x_t \mid f_\theta(x_{t-1}), Q)$, where $f_\theta$ is a deep neural network (DNN). While one can also parameterize the likelihood distribution 2 with a DNN, previous work has shown that making both the likelihood and the dynamics highly expressive can lead to optimization issues (Bowman et al., 2015). Thus, we parameterize the likelihood distribution to be a linear function of $x_t$. Specifically, for spike data, we parameterize equation 2 as $p_\phi(y_t \mid x_t) = \text{Binomial}(y_t \mid 4, \sigma(Cx_t + D))$ where $\sigma$ is the sigmoid function; for real-valued observation, such as behavioral recording, we parameterize equation 2 as $p_\phi(y_t \mid x_t) = \mathcal{N}(y_t \mid Cx_t + D, R)$.

Given a neural time series dataset, $y_{1:T} = [y_1, \ldots y_t, \ldots y_T]$, we are generally interested in inferring the corresponding latent states, $x_{1:T}$, and learning the parameters of the generative model, $\theta$ and $\phi$. Exact inference and learning is difficult as it requires computing the posterior, $p(x_{1:T} \mid y_{1:T})$, and the log marginal likelihood, $p(y_{1:T})$, which are both commonly intractable. We address this challenge by using the seqVAE model—an extension of VAEs for spatio-temporal data (Krishnan et al., 2015). Similar to VAEs, seqVAEs are trained by maximizing a lower-bound of the log-marginal likelihood, commonly referred to as the evidence lower bound (ELBO). Specifically, given data, $y_{1:T}$, the ELBO is defined as

$$\mathcal{L}(y_{1:T}, \theta, \phi, \psi) = \sum_{t=1}^{T} \mathbb{E}_{q_\psi} \left[ \log p_\phi(y_t \mid x_t) + \log p_\theta(x_t \mid x_{t-1}) - q_\psi(x_t \mid y_{1:T}) \right], \quad (3)$$

where $\mathbb{E}_{q_\psi} \equiv \mathbb{E}_{q_\psi(x_{1:T} \mid y_{1:T})}$ and $q_\psi(x_t \mid y_{1:T})$—commonly referred to as the encoder—is a variational approximation to the posterior distribution, $p(x_{1:T} \mid y_{1:T})$. The parameters of the generative model, $\theta, \phi$, and the encoder, $\psi$, are optimized jointly.

While there are various approaches for designing $q_\phi(x_t \mid y_{1:T})$, we follow the parameterization described in Krishnan et al. (2015) for simplicity, where $q_\psi(x_{1:T} \mid y_{1:T}) = \prod_{t=1}^{T} q_\psi(x_t \mid y_{1:T})$ and

$$q_\psi(x_t \mid y_{1:T}) = \mathcal{N}\left( x_t \mid \mu_\psi(y_{1:T}), \sigma_\psi^2(y_{1:T}) \right), \quad (4)$$

where $\mu_\psi(\cdot)$ and $\sigma_\psi^2(\cdot)$ are bidirectional recurrent neural networks.

## 3.2 Alignment of Neural Time Series

Now let's consider a seqVAE model trained on $y_{1:T}$, which we are interested in re-using for a new dataset, $w_{1:T} = [w_1, \ldots, w_T]^1$, where $w_t \in \mathcal{W} \subset \mathbb{R}^{d_w}$. In general, $w_{1:T}$ will not follow a similar distribution to $y_{1:T}$. This can be due to several reasons—there might be drift in the recording probes over sessions, the data might have been collected from a different animal, or using a different recording modality, and so on. The distribution mismatch between $y_{1:T}$ and $w_{1:T}$ prevents straightforward application of the trained seqVAE to $w_{1:T}$.

One approach for re-using this model for $w_{1:T}$ is learning an alignment function between the datasets, $g_\vartheta : \mathcal{W} \to \mathcal{Y}$, that projects $w_t$ to $\mathcal{Y}$, i.e. $\hat{y}_t \equiv g(w_t)$. The projected data can subsequently be fed to the pre-trained encoder, i.e., $q_\psi(x_t \mid g_\vartheta(w_{1:T}))$, $g_\vartheta(w_{1:T}) \equiv [g_\vartheta(w_1), \ldots, g_\vartheta(w_T)]$, thereby enabling us to re-use it for inferring the latent states from $w_{1:T}$. Broadly, the main objective for optimizing the alignment function, $g_\vartheta$, is minimizing the distance between the original data distribution, $p(y_{1:T})$, and the distribution of projected data, $p(\hat{y}_{1:T})$ (Dyer et al., 2017; Duong et al., 2023). Directly minimizing the distance between the two distributions is usually infeasible, as we don't have knowledge about the marginal distributions of the datasets. Moreover, most common distance measures are tractable for a limited class of distributions, many of which are not able to effectively model complex spatio-temporal neural activity.

An alternative is using a supervised learning approach to learn the alignment. Specifically, suppose that we have paired samples from the two datasets, i.e., $\mathcal{D} = \{(w_t, y_t)\}_{t=1}^{T}$. We can then learn $g_\vartheta$ by minimizing the error between $y_t$ and $g_\vartheta(w_t)$, i.e., $\|y_t - g_\vartheta(w_t)\|^2$. Although this approach can recover the optimal alignment, it requires the source dataset $y$, and has the restrictive requirement of paired samples between the datasets.

# 4 Unsupervised Alignment of Neural Time Series

In order to re-use the pre-trained seqVAE, we begin by assuming that the underlying latent dynamics for $w_{1:T}$ are the same as $y_{1:T}$, allowing us to fix the parameters of the trained dynamics model, $p_\theta(x_t \mid x_{t-1})$. This is supported by empirical evidence that the inferred latent dynamics from different neural networks (both biological and artificial) performing the same task are similar (Maheswaranathan et al., 2019; Safaie et al., 2022; Brain et al., 2023). The other components of our pre-trained seqVAE consist of the likelihood function, $p_\phi(y_t \mid x_t)$, and the encoder, $q_\psi(x_{1:T} \mid y_{1:T})$.

---

[1] For ease of presentation, we set the length of $y_{1:T}$ and $w_{1:T}$ to be the same, but the proposed approach does not require this to be the case.

The encoder and likelihood both assume that the observations have a dimensionality of $d_y$ but in general, $d_w \neq d_y$. Moreover, different recording modalities require different likelihoods, thus $p_\phi(\cdot \mid x_t)$ may not be the suitable parametric form for $w_t$. However, given that we paramterize the likelihood as a linear function of $x_t$, re-training a likelihood model for $w_{1:T}$ will result in minimal computational overhead. Thus, we train a new likelihood distribution specific to the new dataset, $w_{1:T}$, i.e., $p_{\phi_w}(w_t \mid x_t)$. In contrast, the encoders for seqVAEs are usually parameterized by large neural networks, such as a bidirectional recurrent neural network, and it would be preferable to keep its parameters fixed. As described in Section 3.2, a way to avoid re-training the encoder is by learning an alignment between the two datasets. In this work, we propose an unsupervised algorithm based off a simple observation.

In the VAE framework, the role of the encoder is to infer the latent states given the observed data, where the optimal encoder corresponds to the true posterior distribution (Blei et al., 2017). Suppose that the pre-trained encoder provides a good approximation to the posterior, i.e., $q_\psi(x_{1:T} \mid y_{1:T}) \approx p(x_{1:T} \mid y_{1:T})$. Intuitively, a good alignment function should facilitate the re-use of the pre-trained encoder to obtain a reasonable approximation to the posterior on the new dataset, i.e., $q_\psi(x_{1:T} \mid g_\vartheta(w_{1:T})) \approx p(x_{1:T} \mid w_{1:T})$. To study the validity of this intuition, we consider a simple linear model that affords analytical tractability; for ease of presentation, we drop the time index. Let $p(x) = \mathcal{N}(0, I)$, $p(y \mid x) = \mathcal{N}(Ax, Q)$, $p(w \mid x) = \mathcal{N}(Cx, R)$ and $g_\vartheta(w) = \vartheta w$, where $I$ is the identity matrix. Based on the previous intuition, we can optimize the parameters of the alignment, $\vartheta$, by minimizing the expected Kullback-Leibler divergence between $q(x \mid g_\vartheta(w))$ and $p(x \mid w)$

$$\vartheta_\star = \arg\min_\vartheta \; \mathbb{E}_{p(w)}\left[\mathbb{D}_{\mathrm{KL}}\left(q\left(x \mid g_\vartheta(w)\right) \| p(x \mid w)\right)\right], \tag{5}$$

which is equivalent to maximizing the expected ELBO

$$\vartheta_\star = \arg\max_\vartheta \; \mathbb{E}_{p(w)}\left[\mathbb{E}_{q(x \mid g_\vartheta(w))}\left[\log p(w \mid x) + \log p(x) - \log q\left(x \mid g_\vartheta(w)\right)\right]\right]. \tag{6}$$

Recalling that the optimal encoder is the posterior—and that the linear and Gaussian model allows for a tractable posterior (Bishop, 2007)—we define $q(x \mid y)$ as

$$q(x \mid y) = p(x \mid y) = \mathcal{N}(\mu(y), \Sigma), \tag{7}$$

$$\mu(y) \triangleq \Sigma A^\top Q^{-1} y, \tag{8}$$

$$\Sigma \triangleq (A^\top Q^{-1} A + I)^{-1}. \tag{9}$$

Thus, $q(x \mid g_\vartheta(w)) = \mathcal{N}(\mu(g_\vartheta(w)), \Sigma)$. The tractability of this simple model allows us to directly compare the solution of equation 6, $\vartheta_\star$, with the optimal alignment with respect to the mean-squared error, $\vartheta_\dagger$

$$\vartheta_\dagger = \arg\min_\vartheta \mathbb{E}_{p(w,y)}\left[(y - \vartheta w)^\top (y - \vartheta w)\right]. \tag{10}$$

In the following proposition, we demonstrate that $\vartheta_\star$ can be expressed as a linear transformation of $\vartheta_\dagger$; the proof can be found in the Appendix (A)

**Proposition 1** *Let $\vartheta_\star$ be the solution of equation 6 and $\vartheta_\dagger$ be the solution of equation 10. Then $\vartheta_\star = \left(I + Q(AA^\top)^{-1}\right)\vartheta_\dagger$, where $I$ is the identity matrix.*

Proposition 1 demonstrates that by optimizing equation 6, we obtain a linear transformation of the optimal alignment, $\vartheta_\dagger$. Moreover, we see that the difference between $\vartheta_\dagger$ and $\vartheta_\star$ is a function of $Q(AA^\top)^{-1}$. Thus, when the new observation noise, $Q$ is small and/or when $AA^\top$ is large, we expect for $\vartheta_\star \approx \vartheta_\dagger$. We emphasize that we are able to implicitly learn a good approximation of the optimal alignment function in an *unsupervised fashion* without paired samples or the source data, $y$.

Inspired by Proposition 1, we move on to designing a general purpose algorithm for unsupervised learning of an alignment function, $g_\vartheta$. A straightforward approach is to jointly learn the parameters of the alignment, $g_\vartheta$, and of the dataset specific likelihood, $p_{\phi_w}(w_t \mid x_t)$, by optimizing the ELBO

$$\mathcal{L}(w_{1:T}, \phi_w, \vartheta) = \sum_{t=1}^{T} \mathbb{E}_{q_{\psi,\vartheta}}\left[\log p_{\phi_w}(w_t \mid x_t) + \log p_\theta(x_t \mid x_{t-1}) - q_\psi\left(x_t \mid g_\vartheta(w_{1:T})\right)\right], \tag{11}$$

where $\mathbb{E}_{q_{\psi,\vartheta}} \equiv \mathbb{E}_{q_\psi(x_{1:T} \mid g_\vartheta(w_{1:T}))}$ and both the latent dynamics, $p_\theta(x_t \mid x_{t-1})$, and the encoder, $q_\psi(x_t \mid y_{1:T})$, are kept fixed.

While optimizing equation 11 is simple and can lead to good empirical performance, we found that it was easy for the optimizer to converge to a suboptimal local minimum. Further investigation revealed that the optimizer would produce latent states that are likely under the one-step ahead dynamics, $\log p_\theta(x_t \mid x_{t-1})$, but would not respect the global dynamics; Fig 4 presents an example (denoted as 1-step prior).

To regularize the optimizer to produce latent states that respect the global dynamics, we replace the one-step ahead dynamics, $\log p_\theta(x_t \mid x_{t-1})$, with a K-step ahead dynamics term, $\sum_{j=1}^{K} \log p_\theta(x_{t-K+j} \mid x_{t-K})$, (Hafner et al., 2019), which encourages the latent states to follow the dynamics over the $k$-step horizon. Following (Hafner et al., 2019), although $\log p_\theta(x_{t-K+j} \mid x_{t-K})$ is intractable, it is straightforward to obtain an unbiased Monte Carlo estimate; in the Appendix B we discuss

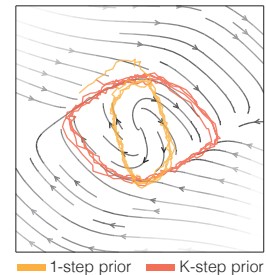

1-step prior    K-step prior

Figure 2: A case for K-step regularization.

how an unbiased estimate is obtained. This leads to the $K$-step ahead ELBO, which we use going forward

$$\mathcal{L}_K(w_{1:T}, \phi_w, \vartheta) = \sum_{t=1}^{T} \mathbb{E}_{q_{\psi,\vartheta}} \left[ \log p_{\phi_w}(w_t \mid x_t) + \sum_{j=1}^{K} \log p_\theta(x_{t-K+j} \mid x_{t-K}) - q_\psi(x_t \mid g_\vartheta(w_{1:T})) \right].$$

We note that, although $\log p_\theta(x_{t-K+j} \mid x_{t-K})$ is intractable, it is straightforward to obtain an unbiased Monte Carlo estimate (Hafner et al., 2019). In Fig 4, we see that using the K-step ahead ELBO leads to much better latents (denoted as $K$-step prior)

## 5 EXPERIMENTS

We validate our alignment approach on synthetic datasets generated with via the Van der Pol dynamics and the Lorenz system. Then, we test our method on neural recordings obtained from the primary motor cortex (M1) in two monkeys during a reaching task (Dyer et al., 2017). We compare the proposed approach against the following methods:

**ERDiff** (Wang et al., 2023). This method uses a pre-trained seqVAE along with a diffusion model with spatio-temporal transformer blocks to estimate the density of latent trajectories on the original dataset $p_s(x_{1:T})$. The alignment function is trained to maximize $\mathbb{E}_{q_{\psi,\vartheta}}[\log p_s(x_{1:T})]$ where the encoder is kept fixed. They additionally optimize a Sinkhorn divergence between the source and target latents.

**NoMAD** (Karpowicz et al., 2022). Given a pre-trained seqVAE, NoMAD fits a multivariate Gaussian to the inferred latent states from the original dataset, $p_y(x) = \mathcal{N}(\mu_y, \Sigma_y)$. The alignment function is trained to maximize $\sum_{t=1}^{T} \mathbb{E}_{q_{\psi,\vartheta}}[\log p_{\phi_w}(w_t)] - \mathbb{D}_{\text{KL}}[p_y(x)\|p_\vartheta(x)]$ where $p_\vartheta = \mathcal{N}(\mu_\vartheta, \Sigma_\vartheta)$ is a Gaussian distribution fit to the latents from the new dataset.

**Cycle-GAN** (Ma et al., 2023). Cycle-GAN leverages adversarial training, via a generative adversarial network, to align new sessions to the original dataset.

**Orthogonal Procrustes** (Schönemann, 1966). An alignment is learned via Orthogonal Procrustes. We note that this requires paired samples from the original and new datasets.

**Re-training**. We train a generative model from scratch on the new dataset as an upper bound on performance.

To isolate the benefits of each method, one seqVAE is trained and is then given to all methods. Due to space constraints, we defer training and architecture details to the Appendix C.

### 5.1 VAN DER POL OSCILLATOR

The Van der Pol oscillator is a two-dimensional nonlinear dynamical system. We consider a noisy version of this system described as follows:

$$\dot{x}_1 = \mu(x_1 - \frac{1}{3}x_1^3 - x_2) + \epsilon, \qquad \dot{x}_2 = \frac{1}{\mu}x_1 + \epsilon, \tag{12}$$

where $\mu = 1.5$ and $\epsilon \sim \mathcal{N}(0, 0.1)$. For training the seqVAE, we generated 1600 trajectories of length $T = 300$ with spike observations where the number of neurons was set to 250, i.e., $d_y = 250$. To avoid aligning the raw spikes (which are high-dimensional), we use a non-linear embedding function that down-projects spikes to 64 dimensions using an MLP before passing it into the encoder.

For evaluating the alignment methods, we generated three more datasets, $w_{1,1:T}$, $w_{2,1:T}$, and $w_{3,1:T}$—each of length $T = 300$—where each dataset has a different number of neurons ($d_{w_1} = 200, d_{w_2} = 250, d_{w_3} = 300$). For the proposed approach, ERDiff and NoMAD, we parameterize $g_\vartheta$ as an MLP. For each dataset, all methods were trained using 500 trajectories and were evaluated on a held-out test set.

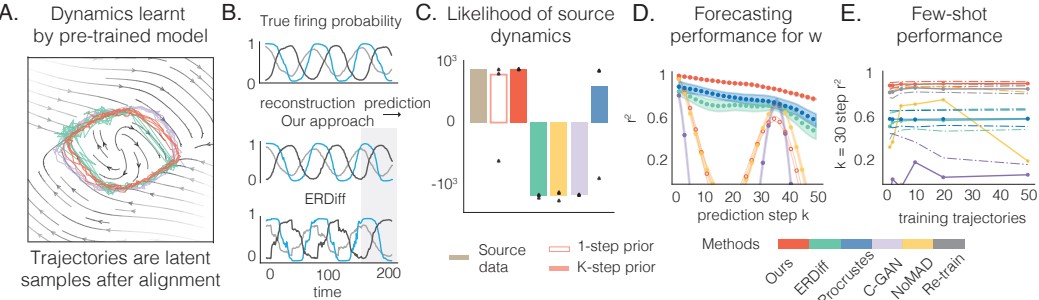

Figure 3: **A.** The vector field is generated from the learnt source dynamics. Sampled latent trajectories from the encoder after aligning the observations using the proposed approach, ERDiff and CycleGAN. **B.** The true firing probability of example neurons on a trial (top) and the reconstruction and prediction for the aligned data using the pre-trained model (below). **C.** Likelihood of source dynamics given inferred latents for the source and aligned data. **D.** K-step prediction $r^2$ performance. **E.** K=30-step prediction performance for various approaches. We plot the median (solid) and the [20, 80] percentile (dotted) $r^2$.

In Fig. 3A, we plot example latent trajectories sampled from the encoder using comparing our approach to ERDiff and CycleGAN. We see that our approach produces smoother latents that respects the pre-trained dynamics; this leads to better reconstructed firing rates and smoother predictions Fig. 3B. To quantify whether the alignment procedures lead to latents that respect the dynamics we compute the likelihood of the inferred latents on the trained dynamics, i.e., $\sum_{t=1}^{T} \mathbb{E}_{q(x_{1:T}|g_\vartheta(w_{1:T}))} \left[ \log p_\theta(x_t|x_{t-1}) \right]$ (Fig. 3C). We see that the proposed method outperforms all comparisons. Moreover, we see that using a $K$-step ahead prior leads to better performance as opposed to the standard 1-step ahead prior.

We subsequently evaluate the methods on their forecasting performance. We use the first 250 time points to infer the latents after aligning and sample 50 steps in the future. We measure the performance by computing the $r^2$ between the true and predicted trajectories (Fig 3D). Our approach performs close to a model trained from scratch on the new dataset. We also test the few-shot performance of these methods. In Fig 3E, we plot the forecasting performance for $k = 30$ as a function of the number of trajectories used for training. Even in the one-shot regime, our method consistently achieves high $r^2$ performance and demonstrates low variance compared to other alignment methods.

To demonstrate that the method can also allow for alignment across recording modalities, we include an experiment in the Appendix where we align real-valued data to the pre-trained model Fig 6. Specifically, $w_t \sim \mathcal{N}(Cx_t, \sigma I)$, where $d_w = 30$ and $\sigma = 0.1$. From Fig. 6, we see that the proposed method performs well and is able to match the forecasting performance of a model trained from scratch.

## 5.2 LORENZ ATTRACTOR

The Lorenz attractor is a three-dimensional system with chaotic dynamics described by the following set of equations,

$$\dot{x_1} = \sigma(x_2 - x_1), \quad \dot{x_2} = x_1(\rho - x_3) - x_2, \quad \dot{x_3} = x_1 x_2 - \beta x_3, \tag{13}$$

where $\sigma = 10$, $\beta = 8/3$, and $\rho = 28$. For training the seqVAE, we generated 1600 trajectories of length $T = 500$ with real-valued observations where $d_y = 40$.

For evaluating the alignment methods, we generated two more datasets, $w_{1,1:T}$, and $w_{2,1:T}$, each of length $T = 500$, where $d_{w_1} = 35$ and $d_{w_2} = 55$. For the proposed approach, ERDiff and NoMAD, we parameterize $g_\vartheta$ as a linear function. For each dataset, all methods were trained using 1,000 trajectories and were evaluated on a held-out test set. We evaluate the models on reconstruction and forecasting, where for forecasting we use 400 time points to infer the latents and sample 50 steps in the future.

In Table 1, we display the reconstruction and forecasting $r^2$ for each of the methods. On reconstruction, we see that both the proposed approach and NoMAD perform very well and are able to match the performance of training a

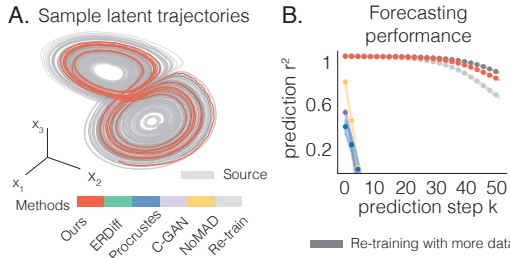

Figure 4: **A.** Latent samples from new data using our approach are aligned with the source latents. **B.** Prediction performance for various alignment methods and re-training a model from scratch.

model from scratch. In contrast, we see that for a prediction horizon of only 5, all the baselines deteriorate. Only the proposed approach is able to match the forecasting performance of a model trained from scratch. In Fig 4, we see that the proposed approach allows for stable forecasting up to 30 time steps ahead.

| Methods | Reconstruction $r^2$ | 5-step ahead $r^2$ |
|---|---|---|
| Re-training | **0.99 ± 0.0005** | **0.99 ± 0.0006** |
| Our approach | **0.99 ± 0.0008** | **0.99 ± 0.0012** |
| ERDiff | -0.08 ± 0.0496 | -0.23 ± 0.0553 |
| NoMAD | **0.99 ± 0.0005** | -0.03 ± 0.0714 |
| Cycle-GAN | 0.86 ± 0.0131 | -0.42 ± 0.0937 |
| Procrustes | 0.68 ± 0.052 | 0.18 ± 0.0833 |

Table 1: Reconstruction and forecasting performance for the Lorenz attractor. The values indicate the median and standard error over the observations from new sessions. We report the prediction performance for 5-step ahead prediction.

Next, we compare the alignment learnt from our approach to the optimal alignment that can be obtained with respect to mean-squared error. In order to do this, we simulated 100 trajectories from the Lorenz attractor, and used the same likelihood models as above to generate observations with paired samples. The alignment function from our unsupervised approach closely matches the optimal (Fig. 7B, RMSE: $0.0017 \pm 0.002$).

## 5.3 NEURAL RECORDINGS

We applied our method to motor cortex recordings from two monkeys (M and C) during a delayed center out reaching task(see (Dyer et al., 2017) for details). Briefly, the monkeys were trained to use a manipulandum to move a cursor to one of eight possible target locations on the screen (Fig. 5A). As they performed the task, electrophysiological activity was recorded from M1 along with the hand position and velocity. For each monkey, two sessions of data were available where the number of correct trials per session ranged from 159 to 215 while the total number of neurons varied from 150 to 167. Following (Wang et al., 2023), we pre-process the data by first binning the neural activity into 20 ms bins. The binned spikes were then smoothed using a 50 ms Gaussian kernel.

We trained a seqVAE on session 1 from monkey M as we observed that the recordings from this session were highly informative about the monkey's behavior relative to the other datasets. We set the latent dimension to be 30 and also learn an embedding that projects the smoothed spikes down to 64 dimensions before being passed into the encoder. To ensure that the latents were also informative of the behavior, we included likelihood terms for both the smoothed spikes and the monkey's hand velocity where a Gaussian likelihood was used in both cases. We treat session 2 from Monkey M along with sessions 1 and 2 from Monkey C as new datasets and use them to investigate the performance of the methods. For the proposed approach, NoMAD and ERDiff, we parameterize

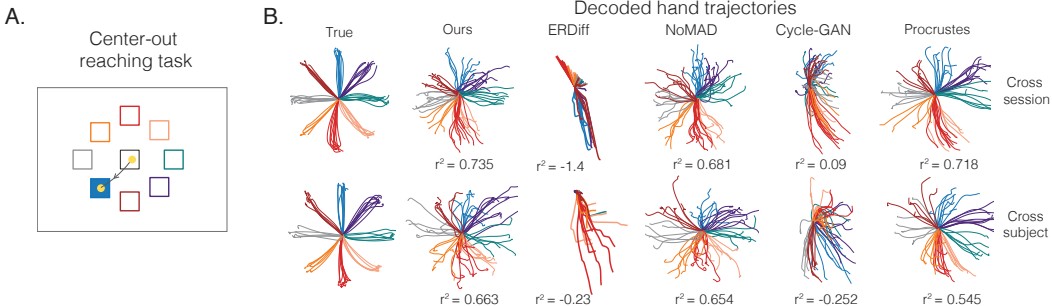

Figure 5: **A.** Schematic depicting the center out reaching tasks that the monkeys performed. **B.** True and decoded monkey hand trajectories after alignment.

$g_\vartheta$ with an MLP. For the proposed approach and NoMAD, the behavior likelihood term was also included in the loss function, but its parameters were kept fixed.

In Fig. 5B, we demonstrate example decoded hand trajectories for some methods where we see that the proposed method, along with NoMAD and Procrustes are able to produce good decoded hand trajectories, both across sessions and across monkeys. In Fig. 8, we plot example latent trajectories where a similar trend is observed. In Table 2, we quantify the reconstruction and forecasting performance for each of the methods. We see that the proposed approach, NoMAD and Procrustes are able to reconstruct the hand trajectories, with NoMAD performing slightly better. In forecasting, we see that only the proposed approach is able to forecast well while the other approaches struggle.

| Methods | Reconstruction $r^2$ | 5-step ahead $r^2$ |
|---|---|---|
| Our approach | $\mathbf{0.66 \pm 0.023}$ | $\mathbf{0.39 \pm 0.071}$ |
| ERDiff | $-0.32 \pm 0.38$ | $-0.23 \pm 0.553$ |
| NoMAD | $\mathbf{0.68 \pm 0.021}$ | $0.15 \pm 0.101$ |
| Cycle-GAN | $-0.15 \pm 0.154$ | $-0.81 \pm 0.121$ |
| Procrustes | $0.61 \pm 0.051$ | $0.07 \pm 0.141$ |

Table 2: Reconstruction and forecasting performance for monkey hand trajectories. Values indicate the median and standard error over the observations from new sessions. We report the prediction performance for 5-step ahead prediction.

## 6 CONCLUSIONS AND LIMITATIONS

In this work, we propose an unsupervised alignment approach that leverages the temporal dynamics learnt from a source dataset to align new data. This enables the re-use of a pre-trained generative model without access to training data or the restrictive requirement of paired samples. We demonstrate the efficacy of our approach by re-using a seqVAE trained on neural recordings from M1 of one monkey to predict behavior on different sessions. This lends further credence to the hypothesis that low dimensional neural representations play a crucial role in neural computation and behavior. The importance of studying these representations in a common space has been previously highlighted (Dabagia et al., 2022) and is naturally afforded by our approach. Moreover, the variability in recordings not explained by the common dynamics assumption can offer a complementary insight into individual differences.

While the proposed approach is promising, there are limitations and room for improvement. Firstly, we assume that we have a good pre-trained model that has learnt the underlying dynamics well. An important direction for future research would be learning the underlying dynamics on a task using multiple datasets to identify generalizable latent representations. Secondly, the proposed approach relies assumes that the latent dynamics are exactly the same across datasets. Thus, we would expect our method to work well on recordings obtained during the same or similar cognitive task as the data used for pre-training. Moreover, this assumption does not take behavioral variability into account. The dynamics on tasks with different structure that require the same computation would introduce additional variability. Fine-tuning the model after aligning recordings is one possibility to get good performance across different contexts and would be an interesting direction for future work.

ACKNOWLEDGMENTS

AV and MP were supported by NIH RF1 DA056404 and the Champalimaud Foundation. We thank the anonymous reviewers for their helpful feedback and comments.

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

## A  PROOF OF PROPOSITION 1

We assume that $p(x) = \mathcal{N}(0, I)$, $p(y \mid x) = \mathcal{N}(Ax, Q)$ and $p(w \mid x) = \mathcal{N}(Cx, R)$. Let $g_\vartheta(x)$ be a linear function of $x$, i.e. $g_\vartheta = \vartheta x$. Using standard Gaussian identities, we can derive the marginal distribution for $y$

$$p(y) = \int p(y \mid x)p(x)dx, \tag{14}$$

$$= \int \mathcal{N}(y \mid Ax, Q)\mathcal{N}(x \mid 0, I)dx, \tag{15}$$

$$= \mathcal{N}(y \mid 0, Q + AA^\top), \tag{16}$$

and $w$

$$p(w) = \int p(w \mid x)p(x)dx, \tag{17}$$

$$= \int \mathcal{N}(w \mid Cx, R)\mathcal{N}(x \mid 0, I)dx, \tag{18}$$

$$= \mathcal{N}(w \mid 0, R + CC^\top). \tag{19}$$

## A.1 DERIVATION OF $\vartheta_\dagger$

We begin by first solving for the parameters of the optimal alignment function by minimizing the expected mean squared error (MSE)

$$\vartheta_\dagger = \arg\min_\vartheta \mathbb{E}_{p(w,y)} \left[ (y - \vartheta w)^\top (y - \vartheta w) \right]. \tag{20}$$

To solve the above optimization, we expand the expected MSE

$$\mathbb{E}_{p(w,y)} \left[ (y - \vartheta w)^\top (y - \vartheta w) \right] = \mathbb{E}_{p(w,y)} \left[ y^\top y + w^\top \vartheta^\top \vartheta w - w^\top \vartheta^\top y \right], \tag{21}$$

$$= \mathbb{E}_{p(y)}[y^\top y] + \mathbb{E}_{p(w)}[w^\top \vartheta^\top \vartheta w] - 2\mathbb{E}_{p(w,y)}[w^\top \vartheta^\top y], \tag{22}$$

$$= \operatorname{tr}(Q + AA^\top) + \operatorname{tr}\left(\vartheta^\top \vartheta(R + CC^\top)\right) - 2\operatorname{tr}(C^\top \vartheta^\top A). \tag{23}$$

To find $\vartheta_\dagger$, we differentiate equation 23

$$2\vartheta(R + CC^\top) - 2AC^\top. \tag{24}$$

We set equation 24 to 0 and solve for $\vartheta_\dagger$

$$2\vartheta_\dagger(R + CC^\top) - 2AC^\top = 0 \tag{25}$$

$$\vartheta_\dagger(R + CC^\top) = AC^\top \tag{26}$$

$$\vartheta_\dagger = AC^\top(R + CC^\top)^{-1}. \tag{27}$$

## A.2 DERIVATION OF $\vartheta_\star$

We now move on to solving for $\vartheta_\star$

$$\vartheta_\star = \arg\max_\vartheta \ \mathbb{E}_{p(w)} \left[ \mathcal{L}(\vartheta, w) \right], \tag{28}$$

$$= \arg\max_\vartheta \ \mathbb{E}_{p(w)} \left[ \mathbb{E}_{q(x|g_\vartheta(w))} \left[ \log p(w \mid x) + \log p(x) - \log q(x \mid g_\vartheta(w)) \right] \right], \tag{29}$$

$$= \arg\max_\vartheta \ \mathbb{E}_{p(w)} \left[ \mathbb{E}_{q(x|g_\vartheta(w))} \left[ \log p(w \mid x) \right] - \mathbb{D}_{\mathrm{KL}} \left[ q(x \mid g_\vartheta(w) \| p(w)) \right] \right], \tag{30}$$

where $\mathbb{D}_{\mathrm{KL}}$ is the Kullback-Leibler divergence. Recalling that the optimal encoder is the posterior, we set $q(x \mid y)$ to be the posterior $p(x \mid y)$

$$q(x \mid y) = p(x \mid y) = \mathcal{N}(\mu(y), \Sigma), \tag{31}$$

$$\mu(y) \triangleq \Sigma A^\top Q^{-1} y, \tag{32}$$

$$\Sigma \triangleq (A^\top Q^{-1} A + I)^{-1}, \tag{33}$$

thus

$$q(x \mid g_\vartheta(w)) = \mathcal{N}(\mu(g_\vartheta(w)), \Sigma), \tag{34}$$

$$\mu(g_\vartheta(w)) \triangleq \Sigma A^\top Q^{-1} \vartheta w, \tag{35}$$

$$\Sigma \triangleq (A^\top Q^{-1} A + I)^{-1}, \tag{36}$$

Due to the linear and Gaussian assumptions placed on the model, we can analytically evaluate $\mathcal{L}(\vartheta, w)$. For conciseness, we use the substitution, $\hat{y} \triangleq \vartheta w$, when appropriate and absorb all constants with respect to $\vartheta$ into $\mathcal{C}$

$$\mathcal{L}(\vartheta, w) = \mathbb{E}_{q(x|g_\vartheta(w))} \left[ \log p(w \mid x) \right] - \mathbb{D}_{\mathrm{KL}} \left[ q(x \mid g_\vartheta(w) \| p(w)) \right], \tag{37}$$

$$= \mathbb{E}_{q(x|g_\vartheta(w))} \left[ \log p(w \mid x) \right] - \frac{1}{2} \mu(\hat{y})^\top \mu(\hat{y}) + \mathcal{C}, \tag{38}$$

$$= \mathbb{E}_{q(x|g_\vartheta(w))} \left[ -\frac{1}{2}(w - Cx)^\top R^{-1}(w - Cx) \right] - \frac{1}{2}\mu(\hat{y})^\top \mu(\hat{y}) + \mathcal{C}, \tag{39}$$

$$= \mathbb{E}_{q(x|g_\vartheta(w))} \left[ w^\top R^{-1} Cx - \frac{1}{2}x^\top C^\top R^{-1} Cx \right] - \frac{1}{2}\mu(\hat{y})^\top \mu(\hat{y}) + \mathcal{C}, \tag{40}$$

$$= w^\top R^{-1} C \mu(\hat{y}) - \frac{1}{2}\mu(\hat{y})^\top C^\top R^{-1} C \mu(\hat{y}) - \frac{1}{2}\mu(\hat{y})^\top \mu(\hat{y}) + \mathcal{C} \tag{41}$$

We now take the expectation over $w$ of the analytical ELBO from equation 41, where to ease notation, we set $T \equiv \Sigma A^\top Q^{-1}$

$$\mathbb{E}_{q(x|g_\vartheta(w))}\left[\mathcal{L}(\vartheta, w)\right] = \mathbb{E}_{q(x|g_\vartheta(w))}\left[w^\top R^{-1} C \mu(\hat{y}) - \frac{1}{2}\mu(\hat{y})^\top C^\top R^{-1} C \mu(\hat{y}) - \frac{1}{2}\mu(\hat{y})^\top \mu(\hat{y})\right] + \mathcal{C}, \tag{42}$$

$$= \mathbb{E}_{q(x|g_\vartheta(w))}\left[w^\top R^{-1} CT\vartheta w - \frac{1}{2}w^\top \vartheta^\top T^\top C^\top R^{-1} CT\vartheta w - \frac{1}{2}w^\top \vartheta^\top T^\top T\vartheta w\right] + \mathcal{C}, \tag{43}$$

$$= \operatorname{tr}\left(R^{-1} CT\vartheta(R + CC^\top)\right) - \frac{1}{2}\operatorname{tr}\left(\vartheta^\top T^\top C^\top R^{-1} CT\vartheta(R + CC^\top)\right) - \frac{1}{2}\operatorname{tr}\left(\vartheta^\top T^\top T\vartheta(R + CC^\top)\right) + \mathcal{C} \tag{44}$$

To find $\vartheta_\star$, we differentiate equation 44 and set it equal to 0.

$$\nabla_\vartheta \mathbb{E}_{p(w)}[\mathcal{L}(\vartheta, w)] = 0, \tag{45}$$

$$T^\top C^\top R^{-1}(R + CC^\top) - T^\top C^\top R^{-1} CT\vartheta(R + CC^\top) - T^\top T\vartheta(R + CC^\top) = 0, \tag{46}$$

$$T^\top C^\top R^{-1} - T^\top C^\top R^{-1} CT\vartheta - T^\top T\vartheta = 0, \tag{47}$$

$$T^\top C^\top R^{-1} CT\vartheta + T^\top T\vartheta = T^\top C^\top R^{-1}, \tag{48}$$

$$C^\top R^{-1} CT\vartheta + T\vartheta = C^\top R^{-1}, \tag{49}$$

$$\left(I + C^\top R^{-1} C\right) T\vartheta = C^\top R^{-1}, \tag{50}$$

$$T\vartheta = \left(I + C^\top R^{-1} C\right)^{-1} C^\top R^{-1} \tag{51}$$

We use the following variant of the Woodbury identity (Petersen et al., 2008) (equation 158)

$$\left(I + C^\top R^{-1} C\right)^{-1} C^\top R^{-1} = C^\top \left(R + CC^\top\right)^{-1}, \tag{52}$$

and plug it into equation 51 to get

$$T\vartheta = C^\top \left(R + CC^\top\right)^{-1}. \tag{53}$$

Recalling that $T \equiv \Sigma A^\top Q^{-1}$ and $\Sigma = \left(A^\top Q^{-1} A + I\right)^{-1}$

$$\Sigma A^\top Q^{-1}\vartheta = C^\top \left(R + CC^\top\right)^{-1}, \tag{54}$$

$$\left(A^\top Q^{-1} A + I\right)^{-1} A^\top Q^{-1}\vartheta = C^\top \left(R + CC^\top\right)^{-1}. \tag{55}$$

Using the same Woodbury identity, we can simplify the above

$$A^\top \left(Q + AA^\top\right)^{-1}\vartheta = C^\top \left(R + CC^\top\right)^{-1}, \tag{56}$$

$$(AA^\top)^{-1} AA^\top \left(Q + AA^\top\right)^{-1}\vartheta = (AA^\top)^{-1} AC^\top \left(R + CC^\top\right)^{-1}, \tag{57}$$

$$\left(Q + AA^\top\right)^{-1}\vartheta = (AA^\top)^{-1} AC^\top \left(R + CC^\top\right)^{-1}, \tag{58}$$

$$\vartheta = \left(Q + AA^\top\right)(AA^\top)^{-1} AC^\top \left(R + CC^\top\right)^{-1}, \tag{59}$$

$$\vartheta = Q(AA^\top)^{-1} AC^\top \left(R + CC^\top\right)^{-1} + AA^\top(AA^\top)^{-1} AC^\top \left(R + CC^\top\right)^{-1}, \tag{60}$$

$$\vartheta = Q(AA^\top)^{-1} AC^\top \left(R + CC^\top\right)^{-1} + AC^\top \left(R + CC^\top\right)^{-1}, \tag{61}$$

$$\vartheta = \left(I + Q(AA^\top)^{-1}\right) AC^\top (R + CC^\top)^{-1}, \tag{62}$$

$$\vartheta_\star = \left(I + Q(AA^\top)^{-1}\right) \vartheta_\dagger \tag{63}$$

## B    EVALUATING K-STEP AHEAD PRIOR

The $K$-step ahead log prior, $\log p_\theta(x_{t+K} \mid x_t)$, is defined as

$$\log p(x_{t+K} \mid x_t) = \int p_\theta(x_{t+1} \mid x_t) \dots p_\theta(x_{t+K-1} \mid x_{t+K-2}) \log p_\theta(x_{t+K} \mid x_{t+K-1}) dx_{t+1:t+K-1} \tag{64}$$

We can rewrite the above as an expectation

$$\log p(x_{t+K} \mid x_t) = \mathbb{E}_{p_\theta(x_{t+1}|x_t)\dots p_\theta(x_{t+K-1}|x_{t+K-2})}\left[\log p_\theta(x_{t+K} \mid x_{t+K-1})\right] \tag{65}$$

which allows for an unbiased estimator by applying the dynamics $K$ times to $x_t$.

# C  ADDITIONAL DETAILS

## C.1  BASELINES

We note that while the proposed approach and NoMAD learn a dataset specific likelihood while aligning, i.e., $\log p_{\phi_w}(w_t \mid x_t)$, the other methods do not. Thus, for the other methods, we first train the alignment function, $g_\vartheta$. Next, we fit $\log p_{\phi_w}(w_t \mid x_t)$ by maximizing $\sum_{t=1}^T \mathbb{E}_{q_\psi(x_{1:T} \mid g_\vartheta(w_{1:T}))} [\log p_{\phi_w}(w_t \mid x_t)]$

## C.2  MODEL ARCHITECTURE AND TRAINING

For all experiments, the seqVAE encoder was parametrized by a bi-directional GRU with 64 hidden units. The latent dynamics were modeled as $p_\theta(x_t \mid x_{t-1}) = \mathcal{N}(f_\theta(x_{t-1}, Q))$ where $f_\theta$ was a two-layer MLP with a width of 256 and tanh activations.

As spike data tends to be very high-dimensional, we avoid learning an alignment in the original space. Instead, when training the seqVAE we also trained an embedding function that projects the spikes down to 64 dimensions before being passed into the encoder. The embedding function was a two-layer MLP with width of 64 and relu activations.

We used Adam to optimize the model. Unless stated otherwise, we used a weight decay of $10^{-4}$ and a learning rate of $1e^{-3}$.

## C.3  EXPERIMENTS

### EVALUATION METRICS AND COMPARISONS

We evaluated all methods on their reconstruction RMSE and the normalized $r^2$. We also report the forecasting performance using the pre-trained model. After aligning the datasets with each method, we use T time points to sample latents from the encoder, $x_{0:T-1}$ and subsequently generate $\hat{x}_{T:T+k}$ using the dynamics. This is decoded to obtain the predicted observations $\hat{y}_{T:T+k}$. The $r_k^2$ is then computed as:

$$r_k^2 = 1 - \frac{\sum_{i=1}^M (y_k - \hat{y}_k)^2}{\sum_{i=1}^M (y_k - \bar{y})^2}$$

where $\bar{y}$ is the mean activity during the trial, and $M$ is the number of testing samples. In the synthetic experiments, we additionally train a likelihood function after learning the alignment for methods that do not optimize for reconstruction. We have not performed an extensive hyperparameter search for the reported methods and instead followed the values recommended in the papers and/or the code.

### ADDITIONAL DETAILS

**Van der Pol oscillator**. We used an embedding function in this experiment with $d_{\bar{y}} = 64$ since the number of simulated neurons for all datasets was over 200. In this case, we directly align to the intermediate embedding except for the approaches that directly work in the observation space (Cycle-GAN, Procrustes). We used an MLP with $dh = 64$ to align new observations in our approach and used a 10-step prior during optimization.

**Lorenz Attractor** In this experiment, we parameterized $g$ as a linear function, and used 1000 trajectories for training all the alignment methods. Due to the chaotic dynamics in the system, we found that using a small, stochastic value the k-step prior was better than using a large k. As a result, we randomly sampled $k = [1, 2]$ during training. We used a weight decay of 1 for the optimizer.

**Motor cortex** When training the reference model from neural recordings with different stimulus conditions, we included the stimulus input when passing data through the embedding function, as well as the prior dynamics. In order to encourage the model to learn smooth dynamics, we randomly jittered the spike counts in $[-2, 2]$ bins during the first half of training. Additionally, the weight decay parameter for the Adam optimizer was set to $10^{-2}$. We did not include the stimulus input when training the alignment function for any method.

# D ADDITIONAL FIGURES

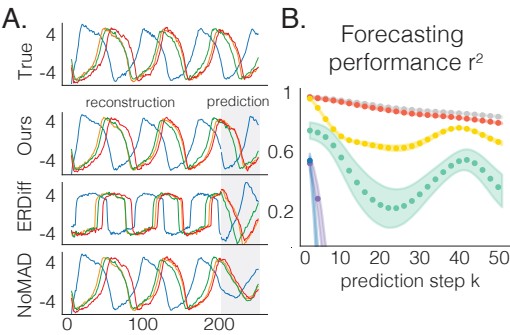

Figure 6: Cross-modal alignment. Aligning Gaussian observations to seqVAE trained on spiking data **A**. Example trajectories of true rates (top) and reconstructed and predicted trajectories after aligning. **B**. k-step prediction performance using the pre-trained model after aligning.

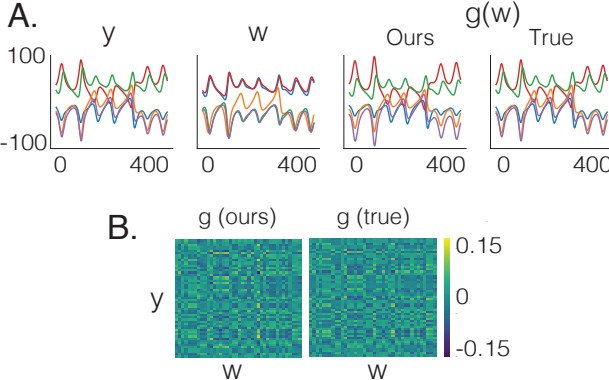

Figure 7: Lorenz attractor with paired samples. **A**. Example units with the same initial condition for $y, w$ and the aligned $g(w)$ obtained using our approach, as well as the optimal alignment. **B**. The optimal $g$ and the one recovered from our method.

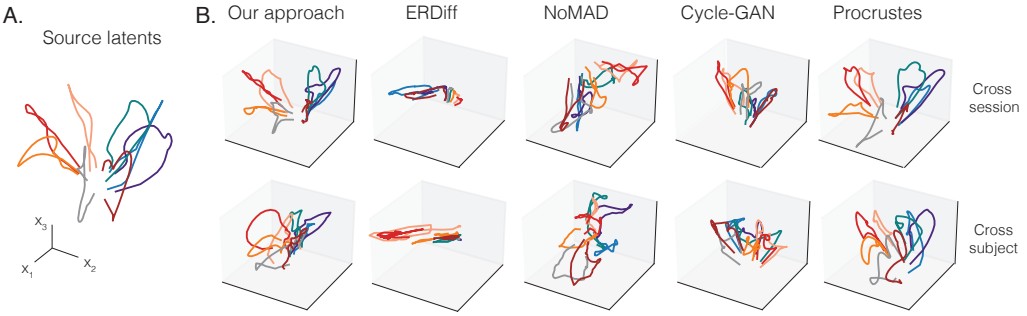

Figure 8: A. Projected latents from the source data used to train the reference seqVAE. Each trajectory corresponds to the stimulus conditioned mean. B. Inferred latents after aligning a session from the same subject (above) and across subject(below) projected on the same space.

