# OpenReview forum: "Leveraging Generative Models for Unsupervised Alignment of Neural Time Series Data"
_ICLR.cc/2024/Conference — ICLR 2024 poster_

### Official Review · Reviewer_nsT1 · 2023-10-14

**Soundness:** 2 fair
**Presentation:** 2 fair
**Contribution:** 2 fair
**Rating:** 6
**Confidence:** 3

**Summary:**

The authors propose an unsupervised alignment approach that can apply a pre-trained model on other neural recordings to the new dataset. They mathematically prove their approach is optimal up to scaler in a simplified case. They validate their methods on a simulation and a real dataset.

**Strengths:**

The authors propose a new unsupervised method to solve the transfer learning problem in neuroscience by re-using the temporal dynamics structure. They also have mathematical proof of the optimality of their method in a simplified case.

**Weaknesses:**

The experiment results are limited. The paper only shows results on 1 real dataset. There are some unclear parts in the paper, which I listed in the Questions section below.

**Questions:**

1. how do you model the latent dynamics pθ(xt|xt−1)? Do you add any smooth constraints? In figure 2b, it seems that the results of the proposed approach are much smoother than ERDiff.
2. what is FA + LDS method that you compare to in figure 2? Can you list a bit more details on how it's implemented?
3. figure 2e, why does the mean k-step r^2 for Re-training decrease as the number of training samples increases? Intuitively the r^2 should increase as there are more training samples.
4. what are the results showing in table 1? Are they results for the real neural dataset? I found the r^2 number in table 1 doesn't match the number in figure 4.
5. in table 1, why the k-step MSE standard deviation is smaller than the stddev of MSE for ERDiff and the proposed method?
6. in figure 4, why the proposed method performs better than ERDiff for poisson observations but not for the smoothed observations, especially Figure 4b (Top, left)?
7. The proposed method assumes that prior dynamics p(xt | xt-1) is fixed, and the encoder q(x | y_embed) is fixed. I wonder if the authors have checked these two assumptions? For example, if you jointly train a model on two datasets and learn the latent dynamics and encoder, will they be similar to the results of training separate models on each dataset?

---

> ### Author Response · Authors · 2023-11-22
>
> We thank the reviewer for their feedback and are in the process of updating the manuscript accordingly. We address specific points below.
> 1. `The experiment results are limited...` In the updated manuscript, our experiments section includes a comparison to three more methods: Procustes, NoMAD [1], and Cycle-GAN [2]. We have also added an experiment based on the Lorenz attractor, a common baseline in the seqVAE literature. In the Appendix, we have included more supplementary experiments such as aligning across recording modalities. Moreover, we emphasize that although we only have data from one experiment, we display results when applying our alignment method on both smoothed spikes, which is a common pre-processing step, and raw spikes.
> 2. `how do you model the latent dynamics pθ(xt|xt−1)? Do you add any smooth constraints? In figure 2b, it seems that the results of the proposed approach are much smoother than ERDiff.` We model our latent dynamics as an MLP with sigmoid activations. We do not apply any smoothness constraint to the latent dynamics. Concerning ERDiff, we note that for all methods we used the same, fixed pre-trained latent dynamics and encoder to avoid confounders. Moreover, ERDiff does not use the latent dynamics for aligning and instead requires training a separate spatio-temporal diffusion model which could be a potential reason for the difference. Thus, the smoothness of our alignment procedure relative to ERDiff is due to our approach, not the dynamics. We will update the manuscript to make this clear.
> 3. `what is FA + LDS method...` We have removed the FA + LDS method as we found it was inappropriate for the experiments. As mentioned above, we have introduced three more comparisons instead.
> 4. `what are the results showing in table 1?` Table 1 demonstrates performance on decoding the hand velocity of the monkey after alignment. The updated manuscript will make this clearer.
> 5. `in table 1, why the k-step MSE standard deviation...` We have updated the table for the monkey reach dataset with new single-step and multi-step metrics for the new comparisons.
> 6. `in figure 4, why the proposed method performs better than ERDiff for poisson observations but not for the smoothed observations, especially Figure 4b (Top, left)?` There was a small error in our evaluation that will be fixed in the updated manuscript.
> 7. `The proposed method assumes that prior dynamics p(xt | xt-1) is fixed, and the encoder q(x | y_embed) is fixed. I wonder if the authors have checked these two assumptions? For example, if you jointly train a model on two datasets and learn the latent dynamics and encoder, will they be similar to the results of training separate models on each dataset?` While this is an interesting experiment, we found that training jointly one generative model for two datasets is highly non-trivial. This is especially true if the datasets are from different modalities. As a proxy, we have compared the performance of using our alignment method to re-training a seqVAE from scratch on the new dataset. We found that the proposed approach performs on par with training from scratch. Moreover, we see that when data is limited, using our alignment approach leads to improvements compared to re-training from scratch. These results will be available in the updated manuscript.
>
> [1] Stabilizing brain-computer interfaces through alignment of latent dynamics. [2] Using adversarial networks to extend brain computer interface decoding accuracy over time.

---

> > ### Author Response · Authors · 2023-11-23
> > **Update**
> >
> > We wanted to inform the reviewer that we have uploaded a revised manuscript.

---

> > > ### Comment · Reviewer_nsT1 · 2023-11-23
> > > **Response to authors' rebuttal**
> > >
> > > Thank the authors for their detailed responses. I keep my current score.

---

### Official Review · Reviewer_H5ee · 2023-10-30

**Soundness:** 4 excellent
**Presentation:** 3 good
**Contribution:** 4 excellent
**Rating:** 8
**Confidence:** 4

**Summary:**

Here, the authors consider the problem posed by the existence of multiple data sets of neural time series from the same or similar tasks, asking whether it is possible to transfer learned latent dynamics from one to the other. They propose a model in which a sequential VAE is first trained on a large data set to establish a latent space ${x}_{1:T}$ (assuming a linear decoder). On a new data set, these dynamics are frozen, a new (linear) decoder is retrained, and the encoder is reused by learning a nonlinear mapping $g(w)$ from the new observations $w$ to the space of the training observations $y$. This requires only the trained encoder and does not require labeled examples. This method is compared with several other alignment proposals, where it produces both quantitative and qualitative improvements.

**Strengths:**

- The problem of aligning multiple data sets across animals and experimental sessions is a major one in neuroscience. An ability to reuse or amortize model training across these would be of significant benefit.
- The method is theoretically well-motivated and fairly flexible. It doesn't appear to require a particular architecture (apart from the linear decoder, which is a limitation of data availability as much as anything).
- The approach appears to produce real qualitative improvements in the learned embedding (Figures 3 and 4).

**Weaknesses:**

- The approach uses a fairly strong assumption that the latent dynamics really are shared across data sets, which all but implies a shared task setup. That is, it doesn't appear to be the case that a sufficiently large task-free data set in one mouse would facilitate embedding of mice performing a task-based behavior. It would be surprising if true, but this weakness should be acknowledged, since this limits the range of applicability.
- Given the large literature on data alignment/domain adaptation both within and without neuroscience, it's a bit surprising that there are only two approaches compared in Table 1.

**Questions:**

- How flexible is this setup to the specific architecture chosen? It's mentioned that using an ELBO that measures log predictive probability several steps ahead is important to achive a good embedding, but it's not entirely clear to me why.
- How complex are the learned dynamics in cases that work versus don't work? The monkey reach data typically have rotational dynamics. Do you see anything more complicated in other data?

---

> ### Author Response · Authors · 2023-11-22
> **Response to reviewer**
>
> We thank the reviewer for their feedback and are in the process of updating the manuscript accordingly. We address specific points below.
> 1. `The approach uses a fairly strong assumption that the latent dynamics really are shared across data sets…` We agree that this is a strong assumption as we would expect that the proposed approach would work well on data from animals doing very similar tasks. We will update the manuscript to acknowledge this weakness.
> 2. `Given the large literature on data alignment/domain adaptation both within and without neuroscience, it's a bit surprising that there are only two approaches compared in Table 1` In the updated manuscript, we have included three more methods as comparisons: the Procrustes method, NoMAD [1] and Cycle-GAN [2].
> 3. `How flexible is this setup to the specific architecture chosen? It's mentioned that using an ELBO that measures log predictive probability several steps ahead is important to achive a good embedding, but it's not entirely clear to me why.` Our method makes no assumption on the architecture of the seqVAE. The only assumption that we make is that the pre-trained seqVAE achieved good performance on the original dataset. Thus, we would expect our method to perform regardless of the architecture. Regarding the k-step ahead prior, we found that during optimization, the standard ELBO used for seqVAE would sometimes get stuck in sub-optimal local minimum. Through further investigation, we found that the alignment procedure would produce latents that would respect the one-step ahead dynamics, $p_\theta(x_t \mid x_{t-1})$, but not the global dynamics. Thus, the $k$-step ahead prior was introduced as a regularizer to ensure that aligned latents would respect the global dynamics. We will update the manuscript to make this clearer.
> 4. `How complex are the learned dynamics in cases that work versus don't work?` Empirically, we found that the method performed well regardless of the complexity. In the updated manuscript, we have included a synthetic experiment based on the Lorenz attractor, a chaotic dynamical system, and found that our proposed method still performs well.
>
> [1] Stabilizing brain-computer interfaces through alignment of latent dynamics. [2] Using adversarial networks to extend brain computer interface decoding accuracy over time.

---

> > ### Comment · Reviewer_H5ee · 2023-11-22
> >
> > I appreciate the authors' responses. I will be maintaining my score.

---

> > > ### Author Response · Authors · 2023-11-23
> > > **Update**
> > >
> > > We wanted to inform the reviewer that we have uploaded a revised manuscript.

---

### Official Review · Reviewer_Jh7W · 2023-10-31

**Soundness:** 3 good
**Presentation:** 3 good
**Contribution:** 3 good
**Rating:** 6
**Confidence:** 3

**Summary:**

The authors present a novel algorithm for leveraging pre-trained seqVAEs for fitting neural recordings. This algorithm builds on the assumption that the neural data in question share the same neural dynamics as the ones used to train the pre-trained model. The two key components to this algorithms are: (1) learn a new observation model from latents to observations and (2) _implicitly_ learn an alignment function from the new observations to the old observations. The authors validated their algorithm on both synthetic data and a monkey reaching dataset.

**Strengths:**

- originality
    To the best of my knowledge this is a new approach for leveraging pre-trained generative models for fitting new data.
- quality & clarity
    - The paper is well-motivated and clearly written, notwithstanding some typos here and there (e.g., missing "In contrast, [our method] does not require..." in last sentence of section 2.)
    - The experiments are relevant and convincing.

**Weaknesses:**

* I am not sure I follow why Proposition 1 implies good alignment? If the space of alignment function that the authors are trying to learn is linear (i.e., $g_\theta(w) = \theta w$), isn't it always the case that any $\theta_\star$ will be some linear offset away from the optimal alignment? Is this offset $B$ supposed to be small somehow?
* The choice of alignment function $g_\theta$ appears crucial to this paper, but there is very little discussion/evaluation about different choices. Specifically, if $g_\theta$ is a point-wise nonlinear function approximator $g_\theta(w_t)$ the authors are implicitly assuming that the latent dynamics are not just qualitatively similar, but **identifcal** between the new and old observations. However, if we make $g_\theta$ to flexible (e.g., another full bi-directional RNN), then we are effectively re-learning the encoder.

**Questions:**

* I assume $q_\phi(x_t, x_{t-1}|y_{1:T}) = q_\phi(x_t|y_{1:T}) q_\phi(x_{t-1}|y_{1:T})$ in equation 3? Might be worth clarifying.
* Is the equation supposed to have $p_\theta(x_t|x_{t-1:t-k})$ instead of $p_\theta(x_t|x_{t-1})$?

---

> ### Author Response · Authors · 2023-11-22
> **Response to reviewer**
>
> We thank the reviewer for their feedback and are in the process of updating the manuscript accordingly. We address specific points below.
> 1. `"I am not sure I follow why Proposition 1 implies good alignment?..."` There was an error in Proposition 1 that will be fixed in the updated manuscript. Namely, the alignment found by the proposed approach, $\vartheta_\star$, can be expressed as a linear transformation of the optimal alignment, $\vartheta_\dagger$, i.e,
> $ \vartheta_\star = \left( I + Q (A A^\top)^{-1}  \right) \vartheta_\dagger ,$
> where $I$ is the identity matrix. From this, we can see that the difference between $\vartheta_\star$ and $\vartheta_\dagger$ is a function of $Q (A A^\top)^{-1}$. Thus for observations that have small noise $Q \approx 0$ and/or small $(A A^\top)^{-1}$ we would expect $\vartheta_\star$ to be similar to $\vartheta_\dagger$. Note, that we can loosely interpret $Q (A A^\top)^{-1}$ as the inverse of the signal-noise-ratio of $w$. Thus, the higher the SNR of $w$ is, the closer $\vartheta_\star$ will be to $\vartheta_\dagger$. The updated manuscript will contain the revised Proposition along with a revised proof as well.
> 2. `"The choice of alignment function appears crucial to this paper,..."` We first note that for experiments that operate on spikes, we parameterize $g_\vartheta$ using a multi-layer perceptron; we will update the manuscript to make this clear. Concerning the expressivity of $g$, we agree that if we allow $g$ to become very expressive then we could effectively re-learn the encoder. Thus, following standard practice in using pre-trained networks, we make $g$ relatively small compared to the generative model. In our experiments, both synthetic and monkey-reach data, we found that our method performed well even when we parameterized $g$ with an MLP. We will update the manuscript to discuss this point as well. We have also included an experiment comparing how the method performs if we set $g$ to be linear as opposed to an MLP.
> 3. `"I assume $q_\phi(x_t, x_{t-1} \mid y_{1:T}) = q_\phi(x_t \mid y_{1:T}) q_\phi(x_{t-1} \mid y_{1:T})$..."` Yes, that is correct! We will clarify this in the manuscript.
> 4. `Is the equation supposed to have $p_\theta(x_t \mid x_{t-k})$ instead of $p_\theta(x_t \mid x_{t-1})$?`. Yes, that is correct! $p_\theta(x_t \mid x_{t-k})$  is the k-step ahead prior that can be expressed as applying the dynamics k times in succession, i.e. $p_\theta(x_t \mid x_{t-k}) = \int  p_\theta(x_{t - k + 1} \mid x_{t - k}) p_\theta(x_{t - k + 2} \mid x_{t - k + 1}) \ldots p(x_t \mid x_{t - 1}) dx_{t - k + 1} \ldots dx_{t - 1} $. We will update the manuscript to make this more clear.

---

> > ### Author Response · Authors · 2023-11-23
> > **Update**
> >
> > We wanted to inform the reviewer that we have uploaded a revised manuscript.

---

> > > ### Comment · Reviewer_Jh7W · 2023-12-01
> > >
> > > I thank the authors' for their clarifications. I will maintain my scores.

---

### Official Review · Reviewer_MYCT · 2023-11-01

**Soundness:** 3 good
**Presentation:** 2 fair
**Contribution:** 3 good
**Rating:** 6
**Confidence:** 4

**Summary:**

Stable and Effective inference models are crucial for decoding neural recordings, yet the need to train new models for each dataset due to variability is computationally demanding and inefficient. This is thus an interesting scientific question. This study introduces a novel alignment method that applies learned dynamics to new data, enabling the reuse of pre-trained models and facilitating the sharing of generative models across different distribution settings. The method's effectiveness is demonstrated by using a seqVAE trained on monkey behavior datasets, underscoring the significance of low-dimensional neural representations and offering a new perspective on handling the neural variability between sessions.

**Strengths:**

1. The paper focuses on an insightful and scientific research question: alignment of neural recordings. Since the generalization ability of models is a great concern.
2. The paper's words and figures are well-written and easy to follow.
3. The proposed method is analytically tractable and is with good theoretical guarantees.

**Weaknesses:**

1. The baselines in the experimental part is too few, just the recently proposed SOTA method ERDiff [1]. There are many classical methods like [2] and [3].
2. The spatio and temporal structure has already been noticed by the SOTA method ERDiff, and ERDiff also employs a generative model (score-based model) for alignment. Thus what's the new motivations and insights of your work?
3. There should be more experiments and empirical results to support your method.

[1] Extraction and Recovery of Spatio-Temporal Structure in Latent Dynamics Alignment with Diffusion Model.
[2] Stabilizing brain-computer interfaces through alignment of latent dynamics.
[3] Robust alignment of cross-session recordings of neural population activity.

**Questions:**

Please consider the things listed in the “Weaknesses” section.
Also please consider providing information regarding any potential future improvements.

---

> ### Author Response · Authors · 2023-11-22
> **Response to reviewer**
>
> We thank the reviewer for their feedback and are in the process of updating the manuscript accordingly. We address specific points below.
> 1. We have updated all of the experiments to include more baselines. Namely, the new baselines are now the Procrustes method, NoMAD [1] and Cycle-GAN [2]. We have also updated the related works section to discuss more methods, including SABLE [3]. We didn’t include SABLE as a comparison as it requires both data and behavior for aligning, making it inapplicable for both synthetic experiments.
> 2. While there are similarities between the proposed approach and ERDiff—specifically, leveraging the spatio-temporal structure and the re-use of generative models—there are also significant differences between the two approaches. Namely, ERDiff requires training a spatio-temporal diffusion model, along with a seqVAE, on the source dataset to perform alignment on a new neural dataset. This adds overhead as if one wants to share their pre-trained seqVAE they must also train a diffusion model. Moreover, ERDiff does not explicitly use the learned latent dynamics when aligning. Instead, only the encoder is re-used and spatio-temporal transformer blocks are used to take into account the spatio-temporal nature of the data. In contrast, our approach is considerably simpler as it only requires training a lightweight alignment function that feeds into the encoder and does not need access to the source dataset. Moreover, the use of the pre-trained latent dynamics is explicitly used as opposed to ERDiff. We provide empirical evidence that—despite its simplicity—our method, which directly uses the pre-trained latent dynamics rather than a spatio-temporal diffusion model, outperforms ERDiff along with the other baselines.
> 3. We have added several experiments to the manuscript, demonstrating the success of the proposed approach. We have included an additional synthetic experiment testing all methods on the Lorenz system, a common benchmark for seqVAEs. We have included experiments demonstrating alignment across modalities (real-valued to spikes and spikes to real-valued).
>
> [1] Stabilizing brain-computer interfaces through alignment of latent dynamics. [2] Using adversarial networks to extend brain computer interface decoding accuracy over time. [3] Robust alignment of cross-session recordings of neural population activity.

---

> ### Comment · Reviewer_MYCT · 2023-11-22
>
> Thank the author for the points in the response. I will hold my original score.

---

> > ### Author Response · Authors · 2023-11-23
> > **Update**
> >
> > We wanted to inform the reviewer that we have uploaded a revised manuscript.

---

### Comment · Area_Chair_PyFy · 2023-11-21
**No discussion?**

Dear authors, dear reviewers,

this is an unusual review process so far: The initial reviews were mostly positive, but some of them asked for concrete clarifications. The authors have, so far, chosen to not provide any clarifications or responses-- we are now one day away from the end of the discussion period.

To the authors: if you want to provide any clarifications, now is the time.
To the reviewers: If there are no clarifications by the authors by the deadline, we will have to discuss the acceptance of the paper base don the information we have so far.

Best, Your AC

---

### Author Response · Authors · 2023-11-22
**Response to all reviewers**

We’d like to thank the reviewers for spending the time to send us feedback and their thoughts on this work. We are currently finishing up updating the manuscript and will post an updated manuscript by tomorrow.

---

### Meta-Review · Area_Chair_PyFy · 2023-12-05

**Metareview:**

The authors consider the problem of aligning multiple similar neural data sets from the same or similar tasks, in order to extract and transfer shared neural dynamics between data sets. They propose a model in which a sequential VAE is first trained on a large data set to establish a latent space with a linear decoder, and retrain  (only) the linear decoder on new data. The compare the method to several alternatives, and report  quantitative and qualitative improvements. Overall, this paper presents a neat application of appropriately chosen machine learning methods to a common problem in neural data analysis.

**Justification For Why Not Higher Score:**

It is a neat paper, but I do not think it is of sufficiently broad interest to be spotlighted.

**Justification For Why Not Lower Score:**

clear consensus for acceptance.

---

### Decision · Program_Chairs · 2024-01-16

Accept (poster)